# Protein-Losing Enteropathy Demonstrated by ^99m^Tc-ASC Lymphoscintigraphy

**DOI:** 10.3390/diagnostics15050583

**Published:** 2025-02-27

**Authors:** Jingnan Wang, Hongli Jing, Fang Li

**Affiliations:** Department of Nuclear Medicine, State Key Laboratory of Complex, Severe, and Rare Diseases, Center for Rare Diseases Research, Beijing Key Laboratory of Molecular Targeted Diagnosis and Therapy in Nuclear Medicine, Peking Union Medical College Hospital, Chinese Academy of Medical Science and Peking Union Medical College, Beijing 100730, China; jingnanwang1991@163.com (J.W.);

**Keywords:** protein-losing enteropathy, ^99m^Tc-ASC, lymphoscintigraphy

## Abstract

A 30-year-old woman presented with progressive edema and mild diarrhea. Laboratory examination revealed hypoalbuminemia. She underwent ^99m^Tc-antimony sulphide colloid (^99m^Tc-ASC) lymphoscintigraphy to evaluate potential loss of protein through gastrointestinal tract caused by lymphatic leakage and detect abnormalities in the lymphatic systems. The images showed abnormal leakage of radiotracers in the bowel, suggestive of protein loss through the gastrointestinal tract. Abnormal visualization of the lower part of thoracic duct and bilateral venous angle was also demonstrated on ^99m^Tc-ASC scintigraphy. It suggested secondary intestinal lymphangiectasis caused by lymphatic obstruction and reflux. Enhanced CT reconstruction of the small intestine revealed roughness and thickening of intestinal wall, consistent with the diagnosis of protein-losing enteropathy.

**Figure 1 diagnostics-15-00583-f001:**
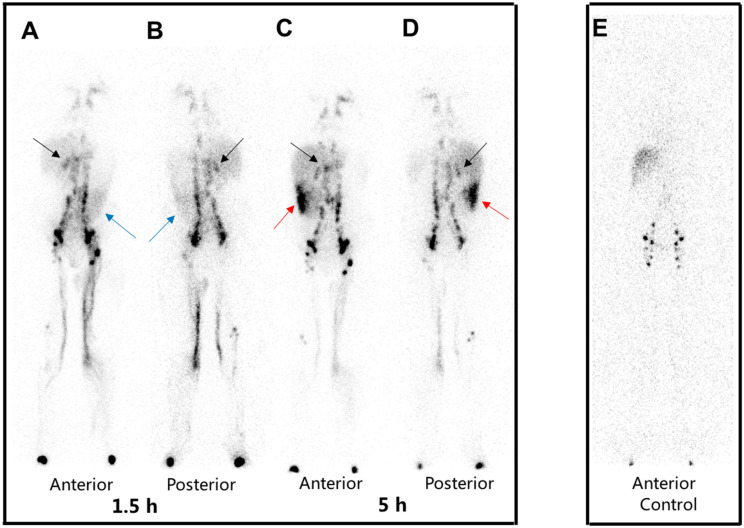
A 30-year-old woman presented with progressive edema from the lower limbs to the face and mild diarrhea for 8 months. Laboratory examination revealed hypoalbuminemia. No abnormalities were found in the examinations for cardiac, liver, and renal function. ^99m^Tc-ASC lymphoscintigraphy was performed to assess clinically suspected potential loss of protein through the gastrointestinal tract caused by lymphatic leakage and detect abnormalities in the lymphatic systems. Whole-body images were acquired at 1.5 h and 5 h after subcutaneous injection of ^99m^Tc-ASC between the 1st and 2nd toe (0.5 mL, 37 MBq/foot), in whole-body scanning mode with a 256 × 1024 matrix at a scan speed of 15 cm/min. The images were visually evaluated, including the distribution, morphology, and uptake function of the lymphatic system. Anterior (**A**) and posterior (**B**) images at 1.5 h showed diffuse radioactivity in the left lower abdominal region (blue arrows) and focal radioactivity in the right mid-upper abdominal region (black arrows). Anterior (**C**) and posterior (**D**) images at 5 h showed notable diffuse activity in the right lower abdominal region (red arrows). The focal radioactivity in the right mid-upper abdominal region (black arrows) was still visible. Bilateral inguinal, iliac, and lumbar trunk lymph nodes were symmetrically visualized with prominent activity. Also noted were bilateral lymph nodes in the mediastinum. Dermal backflow was found in the bilateral lower limbs, consistent with lymphedema. A lymphoscintigraphy image of a control patient is shown. Anterior (**E**) images at 5 h showed that the lymphatic vessels of both lower extremities were clearly visualized with unobstructed flow. The bilateral inguinal lymph nodes, iliac lymph nodes, and lumbar trunks were clearly visualized and symmetrical. The liver and spleen were also visualized. No abnormal radioactive distribution was observed in the abdomen.

**Figure 2 diagnostics-15-00583-f002:**
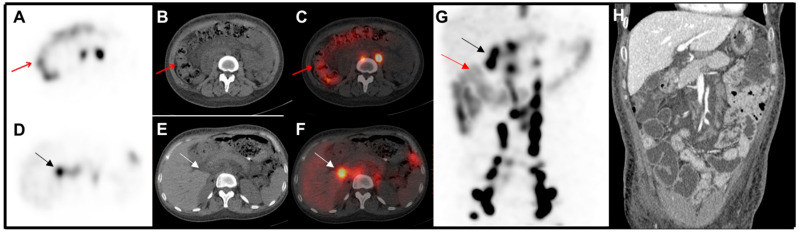
To determine the localization of protein loss, abdominal SPECT/CT was conducted subsequently. Diffused radioactivity was noted in the ascending and transverse colon (red arrows, (**A**) SPECT; (**B**) CT; (**C**) fusion; (**G**) MIP), suggesting anterograde transit of the tracer along the gastrointestinal tract. In addition, the focal radioactivity in the right mid-upper abdominal region (Figure 1, black arrows) demonstrated to be a peripancreatic lymph node on transaxial images (black and white arrows, (**D**) SPECT; (**E**) CT; (**F**) fusion; (**G**) MIP). Visualization of the bilateral venous angle and the lower part of the thoracic duct indicated an outlet obstruction of lymphatic drainage, which caused lymphatic fluid reflux through gastrointestinal tract and lymphedema in the lower extremities in our case. On the enhanced CT reconstruction of the small intestine (**H**), segmentally rough and thickened walls of the descending duodenum and small intestine, as well as a locally narrowed intestinal cavity, were found. These findings indicated the diagnosis of protein-losing enteropathy (PLE). Fecal α-1 antitrypsin clearance was measured in other hospitals and was found to be markedly elevated. Lymphangiography was recommended to identify the lymphatic leakage site and explore potential treatment options. However, the patient refused this procedure and decided to undergo a high-protein and low-fat dietary treatment. PLE is a rare disorder characterized by a severe loss of protein through the gastrointestinal tract, manifesting as symptoms such as edema, diarrhea, and hypoproteinemia, etc. [1,2,3]. The diagnosis of protein-losing gastroenteropathy should be considered in patients with hypoproteinemia in whom other causes, such as malnutrition, heavy proteinuria, and impaired protein synthesis due to liver disease, have been excluded [1]. Nuclear imaging ^99m^Tc-HSA, ^99m^Tc-dextran, and ^99m^Tc-human immunoglobulin scintigraphy have been reported to be useful in diagnosing protein-losing enteropathy [4,5,6,7,8,9,10]. One of the etiologies of protein-losing enteropathy (PLE) is the loss of proteins via lymphatic fluid from the intestine. Pathological mechanisms include conditions such as lymphatic obstruction, congenital abnormalities of the lymphatic system, or increased lymphatic pressure, all of which can result in the leakage of lymphatic fluid into the intestinal tract, contributing to the pathophysiology of PLE. Radionuclide lymphoscintigraphy is a nuclear imaging technique used to visualize the pathways of the lymphatic system, mapping the lymphatic drainage [11,12,13]. This method allows for the identification of abnormalities within the lymphatic system. Specifically, ^99m^Tc-ASC is an ideal agent for lymphoscintigrams because of its small particle size (3–30 micron), which permits early migration into the interstitial space and lymphatics and rapid pickup by lymph nodes. ^99m^Tc-ASC lymphoscintigraphy could reveal abnormalities within the lymphatic systems, enabling the detection of lymphatic leakage in the abdomen. Consequently, this imaging technique provides valuable diagnostic information and insights into the underlying causes of PLE. Lymphoscintigraphy is relatively straightforward to perform and yields specific results. Radiotracers are typically administered via subcutaneous injection and subsequently integrate into the lymphatic system. Nonetheless, lymphoscintigraphy exhibits limitations regarding temporal and spatial resolution, which hinder its ability to localize leakage sites. The radiation exposure should also be considered in clinical application. Reasons for false-negative results may include minimal amounts of lymphatic leakage or insufficient acquisition time, making it undetectable by lymphoscintigraphy. False-positive results, on the other hand, might be due to normal variations in the lymphatic system, leading to misinterpretation of the findings. In summary, our case study suggests that ^99m^Tc-ASC could serve as an effective radiotracer for the diagnosis of PLE, offering a reliable means to identify and assess lymphatic abnormalities underlying PLE.

## Data Availability

No new data were created or analyzed in this study. Data sharing is not applicable to this article.

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
