# Peer review of "Protein-Losing Enteropathy Demonstrated by 99mTc-ASC Lymphoscintigraphy"

_diagnostics, 2025, doi:10.3390/diagnostics15050583_

Round 1
Reviewer 1 Report
Comments and Suggestions for Authors
The paper describes the imaging of a case of Protein-losing gastroenteropathy.
The report is well written, it is easy to read and understand.
I add some comments that may help the authors improve the quality of the text.
Comments:
(1) Could you please add in the text that 99mTc-antimony sulphide colloidan ideal agent for lymphoscintigrams because of small particle size (3-30 micron), which permits early migration into the interstitial space and lymphatics and rapid pickup by lymph nodes.
(2) In the abstract it is stated that " A 30-year-old woman presented with progressive edema and mild diarrhea. Laboratory examination revealed hypoalbuminemia". What was the initial diagnosis and differential diagnosis that led to the decision to perform lymphoscintigraphy?
(3) If protein losing enteropathy was suspected by the measure of albumin and protein levels in blood, was it confirmed with (for example) the presence of alpha-1-antitrypsin in feces?
(4) In Figure 1. Would it be possible to show the expected image of a patient control? The authors have highlighted the areas of interest; I understand that the control patient (healthy) would not show prominent activity.
(5) You may add in the text that "the diagnosis of protein-losing gastroenteropathy should be considered in patients with hypoproteinemia in whom other causes, such as malnutrition, heavy proteinuria, and impaired protein synthesis due to liver disease, have been excluded".
(6) Regarding Figure 2. In case of protein-losing gastroenteropathy, the computed tomography scan or magnetic resonance imaging of the abdomen may demonstrate lymphadenopathy suggestive of lymphoma, mesenteric panniculitis suggestive of sclerosing mesenteritis, and thickening of the bowel wall suggestive of inflammatory bowel disease or infectious colitis. Were these diagnoses exluded?
(7) What was the management of the patient?
Author Response
Respond to the reviewer's comment
The authors are very grateful for the comments. We tried to implement all comments into the manuscript. A detailed answer is given for every comment below.
Comments and Suggestions for Authors
The paper describes the imaging of a case of Protein-losing gastroenteropathy.
The report is well written, it is easy to read and understand.
I add some comments that may help the authors improve the quality of the text.
Comments:
(1) Could you please add in the text that 99mTc-antimony sulphide colloidan ideal agent for lymphoscintigrams because of small particle size (3-30 micron), which permits early migration into the interstitial space and lymphatics and rapid pickup by lymph nodes.
Thanks for this kind comments, we have added in the text (line 72-74).
(2) In the abstract it is stated that " A 30-year-old woman presented with progressive edema and mild diarrhea. Laboratory examination revealed hypoalbuminemia". What was the initial diagnosis and differential diagnosis that led to the decision to perform lymphoscintigraphy?
Thanks for this important question. The patient presented with progressive edema of lower extremities, diarrhea and hypoalbuminemia. No abnormalities were found in the examinations for cardiac, liver and renal function. The manifestation could lead to the suspected loss of protein through gastrointestinal tract in this patient. One of the pathological mechanisms is the loss of protein from the gastrointestinal via the lymphatic fluid. Lymphoscintigraphy can be used to observe whether there is lymphatic leakage and reflect abnormalities of the lymphatic systems, thus provide relevant evidence for the diagnosis and etiology (line 12-13, 25-28).
(3) If protein losing enteropathy was suspected by the measure of albumin and protein levels in blood, was it confirmed with (for example) the presence of alpha-1-antitrypsin in feces?
Yes, the follow-up with the patient revealed fecal α-1 antitrypsin clearance was markedly elevated (line 57-60).
(4) In Figure 1. Would it be possible to show the expected image of a patient control? The authors have highlighted the areas of interest; I understand that the control patient (healthy) would not show prominent activity.
Thanks for this kind suggestion, we have added a patient control in the text (Figure 1).
(5) You may add in the text that "the diagnosis of protein-losing gastroenteropathy should be considered in patients with hypoproteinemia in whom other causes, such as malnutrition, heavy proteinuria, and impaired protein synthesis due to liver disease, have been excluded".
Thanks for this kind suggestion, we have added in the text (line 62-64).
(6) Regarding Figure 2. In case of protein-losing gastroenteropathy, the computed tomography scan or magnetic resonance imaging of the abdomen may demonstrate lymphadenopathy suggestive of lymphoma, mesenteric panniculitis suggestive of sclerosing mesenteritis, and thickening of the bowel wall suggestive of inflammatory bowel disease or infectious colitis. Were these diagnoses exluded?
Thanks for this comment. The CT scan did not find the typical sign of lymphoma or sclerosing mesenteritis. And the colon endoscopy did not find the sign of inflammatory bowel disease or infectious colitis.
(7) What was the management of the patient?
Thanks for this important comment. In this patient, lymphangiography was recommended to identify the lymphatic leakage site and seek potential treatment options. However, the patient refused this procedure and decided to apply a high-protein and low-fat dietary treatment. We have added in the text (line 57-60).
Reviewer 2 Report
Comments and Suggestions for Authors
Dear Authors, your manuscript presents a valuable case study on protein-losing enteropathy (PLE) diagnosed using 99mTc-ASC lymphoscintigraphy. The imaging findings are well-documented, but some areas require improvement for better clarity and scientific impact.
Key Revisions Needed:
Justify the choice of 99mTc-ASC over other nuclear tracers (e.g., 99mTc-HSA, 99mTc-dextran).
Discuss clinical implications: how this imaging technique improve diagnosis and treatment?
Clarify imaging protocol details (acquisition times, tracer dose, diagnostic criteria).
Enhance figure annotations: Add colour-coded overlays or arrows highlighting areas of interest. A schematic representation of the abnormal lymphatic drainage could enhance reader understanding. Clarify the significance of peripancreatic lymph node uptake—is this a secondary finding or clinically relevant?
Limitations of 99mTc-ASC scintigraphy: The manuscript doesn't discuss its limitations, which is essential for scientific balance. Please, including false positives/negatives and their cause, as well as radiation exposure considerations for clinical application.
Update references—Several references are outdated. Please include recent literature (past 5 years).
Your case study is clinically relevant, but these revisions will enhance its impact. Looking forward to your revised manuscript.
Comments on the Quality of English Language
The manuscript is generally well-written and understandable; however, minor grammatical errors and clarity issues should be addressed for improved readability.
Certain phrases are awkwardly worded and would benefit from revision for smoother readability.
A professional language review or proofreading would be beneficial to ensure fluency and clarity.
Round 2
Reviewer 2 Report
Comments and Suggestions for Authors
After reviewing the revised version of the manuscript "Protein-losing enteropathy demonstrated by 99mTc-ASC lymphoscintigraphy", I acknowledge that the authors have adequately addressed the concerns raised in the previous round of review. The revisions have strengthened the manuscript, improving both clarity and scientific rigor.
Based on these improvements, I now consider the manuscript suitable for publication in Diagnostics (ISSN 2075-4418).